



# Global seasonal urban, industrial, and background NO₂ estimated from TROPOMI satellite observations

Vitali Fioletov[1], Chris A. McLinden[1], Debora Griffin[1], Xiaoyi Zhao[1], Henk Eskes[2]

[1]Air Quality Research Division, Environment and Climate Change Canada, Toronto, Canada
[2]Royal Netherlands Meteorological Institute, De Bilt, the Netherlands

*Correspondence to*: Vitali Fioletov (Vitali.Fioletov@outlook.com or Vitali.Fioletov@ec.gc.ca)

**Abstract**. The tropospheric $NO_2$ vertical column density (VCD) values measured by the Tropospheric Monitoring Instrument (TROPOMI) were used to study the $NO_2$ variability and estimate urban $NO_x$ emissions for 261 major cities worldwide. The used algorithm isolated three components in tropospheric $NO_2$ data: background $NO_2$, $NO_2$ from urban sources, and from industrial point sources, and then each of these components was analysed separately. The method is based on fitting satellite data by a statistical model with empirical plume dispersion functions driven by a meteorological reanalysis. Unlike other similar studies that studied plumes from emission point sources, this study included the background component as a function of the elevation in the analysis and separated urban emissions from emissions from industrial point sources. Population density and surface elevation data as well as coordinates of industrial sources were used in the analysis. The largest per capita emissions were found at the Middle East and the smallest were in India and South Africa. The largest background component was observed over China and parts of Europe, while the smallest was over South America, Australia, and New Zealand. Differences between workday and weekend emissions were also studied. Urban emissions on Sundays (or Fridays for some countries) are typically 20%-50% less than workday emissions for all regions except China. The background component typically does not show any significant differences between workdays and weekends suggesting that background $NO_2$ has a substantially longer lifetime compared to that in the urban and industrial plumes.





## 1 Introduction

Nitrogen oxides ($NO_x$, taken here to be nitric oxide (NO) and nitrogen dioxide ($NO_2$)) are major air pollutants whose emissions are regulated in many countries. They originate from various anthropogenic (fuel combustion) and natural (e.g., biomass burning, lightning) sources. $NO_x$ from the combustion of fossil fuels is generally in the form of NO, but is oxidized rapidly forming a pseudo steady-state with $NO_2$. $NO_2$ is linked to respiratory health issues (Health Canada, 2018) and has negative environmental impacts such as acid rains (Burns et al., 2016).

Satellite measurements of one component of $NO_x$, $NO_2$ have a long history. Satellite observations of tropospheric $NO_2$ columns began with the nadir-viewing GOME (Global Ozone Monitoring Experiment) in 1996 (Martin et al., 2002) with several successors such as OMI (Ozone Monitoring Instrument) (Duncan et al., 2015; Krotkov et al., 2016; Lamsal et al., 2015, 2021; Levelt et al., 2018) and TROPOMI (Tropospheric Monitoring Instrument) (van Geffen et al., 2020, 2022; Veefkind et al., 2012). Most recently $NO_2$ measurements become available from geostationary satellite missions such as operational Geostationary Environment Monitoring Spectrometer (GEMS) (J. Kim et al., 2020, Seo et al., 2024), and Tropospheric emissions: Monitoring of pollution (TEMPO) (Zoogman et al., 2016). Another geostationary mission, Sentinel 4 (Stark, 2013), is scheduled for 2024.

These satellite instruments provide measurements of tropospheric $NO_2$ vertical column density (VCD), a geophysical quantity representing the total number of molecules (or total mass) per unit of area in the troposphere. Due to its relatively short lifetime, a few hours within a plume during the day, $NO_2$ is elevated near sources such as urban areas (Beirle et al., 2019; Lorente et al., 2019; Lu et al., 2015) and industrial locations such as power plants and oil refineries (Liu et al., 2016; McLinden et al., 2012). Satellite data have been used to better understand $NO_x$ sources, sinks, distributions, and trends (Beirle et al., 2011, 2019; Goldberg et al., 2021b; Liu et al., 2016; Lorente et al., 2019; Lu et al., 2015; Martin et al., 2002; McLinden et al., 2012; Stavrakou et al., 2020; Vîrghileanu et al., 2020) as well as to estimate $NO_x$ emissions. Several methods have been developed for such emission estimates (Streets et al., 2013): inverse modelling (Konovalov et al., 2006; Mijling and van Der A, 2012), flux divergence (Beirle et al., 2019; 2021) as well as methods based on a rotation of satellite $NO_2$ pixels around the source and then fitting the plume by one-dimensional (Lange et al., 2022; Pommier et al., 2013) or two-dimensional exponentially modified Gaussian (EMG) function (Fioletov et al., 2022, McLinden et al., 2020).

Tropospheric VCDs, together with surface, in-situ $NO_2$ measurements, were both observed to decline during the COVID-19 lockdown, first in China and then worldwide (Bao and Zhang, 2020; Bauwens et al., 2020; Ding et al., 2020; Gkatzelis et al., 2021; Kanniah et al., 2020; Keller et al., 2021; Koukouli et al., 2021; Liu et al., 2020; Vadrevu et al., 2020; Vîrghileanu et al., 2020; Zhang et al., 2021). This decline was observed all over the world: in the U.S. and Canada (Bauwens et al., 2020; Goldberg et al., 2020; Griffin et al., 2020), Europe (e.g., Bar et al., 2021; Barré et al., 2021), India (Mirsa et al., 2021; Hassan et al., 2021), Pakistan (Ghaffar et al., 2021; Mehmood et al., 2021), Brazil (Dantas et al., 2020; Siciliano et al., 2020), and other countries (Ass et al., 2020; Aydin et al., 2020; Fu et al., 2020). More information about the COVID-19 restrictions on atmospheric pollutants can be found in overview papers (Gkatzelis et al., 2021; Levelt et al., 2021). Satellite



data also demonstrated a decline in emissions by comparing the NO$_2$ emissions estimates before and after the lockdown started (Lange et al., 2022; Fioletov et al., 2022).

The concentration of air pollutants over large cities, including NO$_2$, is different from weekends to weekdays due to reduced industrial activity and traffic (Cleveland et al., 1974; Elkus and Wilson, 1977). This is known as the "weekend effect".

This effect was intensively studied using ground-based (Butenhoff et al., 2015; Domínguez-López et al., 2014; Khoder, 2009; Nishanth et al., 2012) and satellite data (Beirle et al., 2003; Goldberg et al., 2021a; Jeong and Hong, 2021; Kaynak et al., 2009; Stavrakou et al., 2020). The estimated amplitude of the workday-weekend difference is about 20%-40% (Goldberg et al., 2021a; Murphy et al., 2007), although it is different from city to city (Lange et al., 2022). Unlike large cities, rural areas with other predominant sources of NO$_x$ (soil emissions, biomass burning, lightning) show no indication for a weekly NO$_x$ cycle.

Satellite NO$_2$ observations also do not show any weekly pattern in rural areas (Kaynak et al., 2009).

In this study, an algorithm previously developed to estimate the COVID-19 lockdown impact on tropospheric NO$_2$ over major urban areas (Fioletov et al., 2022) was applied to the available 2018-2023 TROPOMI data to estimate urban and industrial emissions as well as the background NO$_2$ distribution for four seasons. Emissions derived using this algorithm demonstrated good agreement with the reported industrial emissions from the US power plants (Fioletov et al., 2022) and

urban emission estimates by Lange et al. (2022). The algorithm is based on a multisource dispersion function fitting approach, originally developed to estimate emissions from sulfur dioxide (SO$_2$) point and area sources (Fioletov et al., 2017; McLinden et al., 2020). The approach is based on fitting TROPOMI measurements by statistical models with empirical plume dispersion functions driven by a meteorological reanalysis. The analysis was done using data over 3° by 4° areas around major cities. The statistical models included three components related to (1) plumes from urban sources, (2) plumes from industrial point

sources, and (3) background NO$_2$. The parameters of the statistical model link the satellite NO$_2$ values to proxies related to elevation and population density as well as to locations of large industrial point sources. The algorithm estimates NO$_2$ mass and derives emissions, while the actual emissions are typically in the form of NO that rapidly reacts with ozone producing NO$_2$. Although the emission estimates were done for NO$_2$, they can be upscaled to derive total NO$_x$ emissions. We followed the approach used by Beirle et al., 2021 and Lange et al., 2022 for upscaling NO$_2$ estimates to derive also total NO$_x$ emissions.

This paper is organized as follows: Section 2 describes various data sets used in the study; the statistical models used in this study are discussed in Section 3. In Section 4, emission estimates for individual urban areas and for large regions are discussed. The weekend effect is discussed in Section 5 and changes in NO$_2$ emissions are described in Section 6. Discussion and conclusions are given in Section 7. The upscaling of NO$_2$ to NO$_x$ emissions is described in the Appendix.

## 2 Data Sets

### 2.1 TROPOMI NO$_2$ VCD data

The TROPOMI instrument is a space-borne, nadir-viewing, imaging spectrometer, onboard of the European Space Agency (ESA) and EU Copernicus Sentinel 5 Precursor (S5p) satellite, was launched on 13 October 2017 (van Geffen et al., 2022;



Veefkind et al., 2012). It measures in the ultraviolet and visible (270–500 nm), near-infrared (675–775 nm) and shortwave infrared (2305–2385 nm) spectral bands. The satellite is in a Sun-synchronous, low-Earth (825 km) orbit with a daily equator crossing time of approximately 13:30 local solar time and the swath width of 2,600 km (van Geffen et al., 2018). At nadir, the instrument has a high spatial resolution of $3.5 \times 7$ km$^2$ at the beginning of operation and that was further reduced to $3.5 \times 5.6$

km$^2$ on 6 August 2019. To obtain tropospheric NO$_2$ VCD, the stratospheric portion of the total NO$_2$ column is subtracted using a global model estimate that is refined using data assimilation (Boersma et al, 2004). Our analysis is based on version 2 level 2 TROPOMI tropospheric NO$_2$ VCD data that are available from the Copernicus open data access hub (https://dataspace.copernicus.eu accessed on June 22, 2024). The NO$_2$ data used were PAL, v 2.3.1 (until July 2022) and offline mode (OFFL) v2.4 and 2.5 (end of July 2022- November 2023). These versions were available at the time of this study. A

recent analysis by Nawaz et al. (2024) suggested that v 2.4 introduces an artificial step change of about 5-10% lower NO$_2$ in an urban area that may affect our estimates of long-term changes (Section 6). The main difference between versions 2.3.1 and 2.4 is the use in 2.4 of a higher resolution directional Lambertian equivalent reflectivity (DLER) climatology and a higher resolution of a priori NO$_2$ profiles (Nawaz et al., 2024). Since most of our results represent characteristics integrated over 3° by 4° areas around major cities, the impact of the change of the version should be smaller than 5-10% step change mentioned

above.  Only data with the quality assurance value (*qa_value*) higher than 0.75 (van Geffen et al., 2018) were used. A *qa_value* of 0.75 removes problematic retrievals, errors, and partially snow/ice-covered scenes. In addition, satellite pixels with a solar zenith angle greater than 75 degrees and with cloud radiance fraction above 0.3 were excluded from the analysis. TROPOMI NO$_2$ VCD values represent the total number of molecules per unit area below tropopause and are often given in molecules or moles (one mole is equal to $6.022 \times 10^{23}$ molecules) per square metre or centimetre as well as in Dobson Units (DU, 1 DU =

$2.69 \times 10^{16}$ molec cm$^{-2}$).  The specified random uncertainty of a single TROPOMI tropospheric NO$_2$ VCD measurement is about $5 \times 10^{14}$ molec cm$^{-2}$ (or 0.026 DU) (ESA EOP-GMQ, 2017; van Geffen et al., 2020). Since TROPOMI has only one daily overpass at most locations, diurnal NO$_2$ variations may affect satellite-based emission estimates. Unlike surface concentrations, the diurnal variations of NO$_2$ VCDs are relatively small (Herman et al., 2009; Chong et al., 2018). However, since we do not have information about night-time NO$_2$ VCDs, the presented results are limited to daytime emissions only.

**2.2 Wind data**

As in previous studies (Fioletov et al., 2022, 2015; McLinden et al., 2020; Zoogman et al., 2016), the emission estimation algorithm is based on the plume dispersion function that uses the wind speed and direction obtained from European Centre for Medium-Range Weather Forecasts (ECMWF) ERA5 reanalysis data (C3S, 2017; Dee et al., 2011; Hersbach et al., 2020). The wind speed and direction from the reanalysis data were merged with the tropospheric NO$_2$ value for each TROPOMI pixel.

The reanalysis wind data have one-hour temporal resolution and are available on a 0.25° horizontal grid. U- and V- (west-east and south-north, respectively) wind-speed components were then linearly interpolated to the location of the centre of each TROPOMI pixel and to overpass time.  The ERA5 wind components at 1000, 950, and 900 hPa were averaged to obtain the used wind value (that approximately corresponds to the mean winds between 0 and 1 km). The results are not very sensitive



to the wind profile within this range because the boundary layer wind is relatively constant (Beirle et al., 2011; Zhao et al., 2024) except close to the surface. Note that in ERA5 reanalysis in pressure co-ordinates, when the surface pressure is smaller than that at a given level (e.g., 1000 hPa) the values will simply duplicate the winds at the lowest pressure available.

## 2.3 Population density data, city selection, and elevation data

The Gridded Population of the World (GPW) dataset (SEDAC, 2017) was used as a source for of the population density data. GPW data are on 0.042 degree (2.5 arc-minute) grid and consists of estimates of number of persons per square kilometre based on counts consistent with national censuses and population registers. When coarser resolution data were required, they were obtained by averaging the original data within the new grid cells.

The analysis was performed for the same 261 urban areas around the world as in Fioletov et al., 2022. Information
about large city location and population was obtained from the World Cities Database available from https://simplemaps.com/data/world-cities (accessed on May 10, 2021). All cities with population greater than 1 million were considered. As in the previous study (Fioletov et al., 2022), we also included several European national capitals with population between 700 thousand and 1 million. For China, we raised the limit and considered only cities with population greater than 6 million to keep the number of analysed areas similar to other regions. The cities were grouped into 14 regions and regional
characteristics were calculated by averaging estimates for individual cities within the region.

In the absence of major sources, tropospheric $NO_2$ VCD depends on the thickness of the troposphere that is affected by the elevation. Elevation data from 2-Minute Gridded Global Relief Data (ETOPO2v2) database (NOAA, 2006) were used as a proxy to estimate the background component in the statistical models. When lower resolution data were required, they were obtained by averaging the original data within the new grid cells.
Information about workdays and weekends in different countries was obtain from the "Time and Date AS" web site (Thorsen, 2024; https://www.timeanddate.com/calendar/ accessed on Mar 15, 2024).

## 2.4 Industrial point source locations

Coordinates of industrial point sources are required by the used algorithm. We used the same list of point sources as in Fioletov et al., (2022). For the U.S., information about the sources was taken from the U.S. Environmental Protection Agency (EPA)
National Emissions Inventory (NEI) (EPA, 2020) and from eGRID database (https://www.epa.gov/egrid/download-data, accessed on April 5, 2024) for 2018 and 2019 were used. For Canada, the sources coordinates were from the Canadian National Pollutant Release Inventory (NPRI, 2020) are used. Only Canadian and U.S. sources with annual emissions greater than 0.5 kt of $NO_x$ per year were selected and used in this study. Coordinates of the European industrial point sources were obtained from European Industrial Emissions Portal (https://industry.eea.europa.eu/) (accessed on March 20, 2024). Only sources that
emitted more than 0.5 kt yr$^{-1}$ of $NO_x$ are included in the analysis.



For other regions, three sources of information on industrial source location were used. (1) The world powerplant database (https://globalenergymonitor.org/projects/global-coal-plant-tracker/) (accessed on March 20, 2024) was used to find locations of power plants. (2) Oil and gas-related industrial factories and other sources were also obtained from the $SO_2$ emission source catalogue (Fioletov et al., 2023). (3) Missing sources were also added based on the analysis of the $NO_2$ residuals maps and then confirmed using satellite imagery as discussed in Fioletov et al., (2022).

## 3 The fitting algorithms and statistical models

The technique used here is based on the approach from our previous study (Fioletov et al., 2022) that is briefly described below. All satellite measurements over an 3° by 4° areas (roughly, 330 km by 330 km at 42°N) around large cities taken during a certain period are linked to locations of industrial point sources as well as to population density and elevation-related proxies by a statistical (linear regression) model. The predictor functions of the statistical model represent plumes from the known industrial sources and population grid emission strengths are the unknown parameters of the model. In addition, the statistical model includes a term that links the elevation data with the background $NO_2$ distribution. The parameters of the model were estimated by the least squares method using all data collected during 3-month periods in different years. The following statistical model was used by Fioletov et al., (2022):

$$TROPOMI\ NO_2 = \alpha_0 + (\beta_0 + \beta_1(\theta - \theta_0) + \beta_2(\varphi - \varphi_0)) \cdot exp(-H/H_0) + \alpha_p \Omega_p + \Sigma\ \alpha_i \Omega_i + \varepsilon \qquad (1)$$

where $\alpha_0$, $\alpha_p$, $\alpha_i$, $\beta_0$, $\beta_1$, and $\beta_2$ are the unknown regression parameters of the statistical model; $H$ is the elevation above sea level and the empirical scaling factor $H_0$=1.0 km was introduced to make the exponential argument dimensionless and to account for altitudinal dependence better; and $\varepsilon$ is the residual noise. Plumes are described by $\Omega_p$ and $\Omega_i$ functions for the population density-related distributed source (or area source) and for industrial point sources respectively. The plume function for an industrial source $i$ has a form $\Omega_i = \Omega (\theta, \varphi, \omega, s, \theta_i, \varphi_i)$ where $\theta$ and $\varphi$ are the satellite pixel coordinates; $\omega$ and $s$ are the wind direction and speed for that pixel on that day and time; and $\theta_i$ and $\varphi_i$ are the source coordinates.

The $\alpha_0 + (\beta_0 + \beta_1(\theta - \theta_0) + \beta_2(\varphi - \varphi_0)) \cdot exp(-H/H_0)$ term (four fitted coefficients) represents the mean background tropospheric $NO_2$ distribution around the location with coordinated $\theta_0$ and $\varphi_0$. It is assumed that that term to be declining exponentially with elevation, i.e., within the analyzed 3° by 4° area, the higher is the elevation the lower the background tropospheric $NO_2$ VCD is. It was also assumed that this contribution from elevation depends on geographical coordinates only and not on the winds. The $\alpha_0$ parameter was added to the model to account for remaining free-tropospheric $NO_2$ at high elevations where $exp(-H/H_0)$ is very close to 0.

The $a_p \Omega_p$ term represents urban emissions and the composite plume function $\Omega_p$ is a sum of plume functions of all individual cell centres multiplied by the grid cell population. It was assumed that emissions from each grid cell are proportional to the cell population and the coefficient of proportionality is the same for the entire analysed area. Thus, we just need to estimate one coefficient ($\alpha_p$) that is proportional to the annual emissions per capita. A 0.2° by 0.2° population density grid was





used in the analysis and original population density data were converted to that grid by averaging population density data within each grid cell.

The $\Sigma\ \alpha_i\Omega_i$ term (variable number of coefficients $\alpha_i$ from zero to few dozens) reflects contributions of plumes from individual industrial point sources. An unknown parameter ($\alpha_i$) represents the total $NO_2$ mass emitted from the point source $i$. On a given day, $\Omega_i$ is a contribution to tropospheric $NO_2$ VCD over the location with coordinates $\theta$ and $\varphi$ from a source that contributes one unit of $NO_2$ mass to total $NO_2$ around that source. It was restricted to $\alpha_i \geq 0$. Accounting this plume contribution makes estimates of urban emissions more accurate. Since this study is focused on urban emissions and since the contribution of industrial sources is very different from one urban area to another, industrial emissions are not discussed in detail. If an emission source is located within an urban area, it could be difficult to separate its emissions from urban emissions since $\Omega_p$ and $\Omega_i$ functions would be correlated. For this reason, industrial point sources located in the 0.2° by 0.2° cells where the population is greater than 600,000 people were excluded. This is an empirically estimated limit, and, in a few cases of very large cities (New York, Moscow), it was manually adjusted. Note that for point sources located in close proximity, their plume functions $\Omega_i$ could be highly correlated. In such cases, emissions from individual industrial sources often cannot be estimated. Fioletov et al., (2022) suggested an algorithm to group the sources into independent clusters, so total emissions from such clusters can be estimated. We used that algorithm in this study, although it is not necessary if only total emissions from all point sources in the area are estimated to separate them from urban emissions or if all industrial sources are isolated remote sources.

The plume functions $\Omega$ are EMG functions that are commonly used to approximate plumes of VCDs of trace gases such as $NO_2$, $SO_2$, and ammonia (Beirle et al., 2011, 2014; Dammers et al., 2019; Fioletov et al., 2017, 2015; de Foy et al., 2015; Liu et al., 2016; McLinden et al., 2020). See Fioletov et al., (2022), their Appendix A for details. The plume functions $\Omega$ depends on two prescribed parameters, the lifetime ($\tau$) reflects the rate at which $NO_2$ is removed from the plume due to chemical conversion or physical removal such as deposition, and the plume width ($w$) that largely depends on the satellite pixel size. The same values of $w=8$ km as in Fioletov et al., (2022) is used and the value of $\tau$ will be discussed later.

The parameters $\alpha_0$, $\alpha_p$, $\alpha_i$, $\beta_0$, $\beta_1$, and $\beta_2$ were estimated from the fit of TROPOMI data using the statistical model Eq.1 The fitting was done for all satellite pixels centered within 3° by 4° areas around large cities and collected during the analyzed period by minimization of the squares of the residuals ($\varepsilon$).

The model parameters $\alpha_p$ and $\alpha_i$ represent the total $NO_2$ mass emitted from the population grid and from the source $i$. respectively. The lifetime $\tau$ determines the emission rate ($E$): for an industrial point source $i$ can be expressed as $E_i = \alpha_i/\tau$. Similarly, the urban emission rate is $E_p = \alpha_p/\tau$. Although seasonal emissions were calculated, emissions rates were expressed in kt y$^{-1}$ to make it easier to compare with available annual emissions inventories.

In this study, three different variants of the statistical model discussed above were used. Model 1 was the model given by Eq.(1). The fitting was done for each season for years from 2018 to 2023. This model was used to estimate the evolution of the urban and background components over time. The Model 2 was developed to estimate the mean emissions and $NO_2$ distribution and study the workday vs. weekend effect, and it had the form:



$$TROPOMI\ NO_2 = (\alpha_0 + (\beta_0 + \beta_1(\theta - \theta_0) + \beta_2(\varphi - \varphi_0))\cdot exp(-H/H_0)) \cdot(1+\gamma_{11}I_1+\gamma_{12}I_2+\gamma_3I_{13})$$

$$+ \alpha_p \Omega_p\cdot(1+\gamma_{21}I_1+\gamma_{22}I_2+\gamma_{23}I_3) + \Sigma\ \alpha_i\Omega_i\cdot(1+\gamma_{31}I_1+\gamma_2I_{32}+\gamma_3I_{23}) + \varepsilon \qquad (2)$$

where $I_1$ is an indicator function for Friday: it equals 1 if for Friday measurements and 0 otherwise. Similarly, $I_2$ and $I_3$ are indicator functions for Saturday and Sunday and $\gamma_{2j}$ are the regression model parameters that represent the departure of

Friday's-Sunday's characteristics from these on workdays. In other words, the urban component term is $\alpha_p\Omega_p$ when measurements taken workday are fit and $\alpha_p\ \Omega_p\cdot(1+\gamma_{21})$ for measurements on Friday (and similar terms for Saturday and Sunday). The fitting was done for each month of the year using all available years, i.e., it has 12 sets of parameters. This model estimates the average urban and industrial emissions and background component and does not include terms that depend individual years. As Model 2 is focused on typical (mean) characteristics, the period affected by the COVID-19 restriction

(March-June 2020) was excluded from the analysis.

The Model 3 was focused on estimating changes of urban emissions over time:

$$TROPOMI\ NO_2 = (\alpha_0 + (\beta_0 + \beta_1(\theta - \theta_0) + \beta_2(\varphi - \varphi_0))\cdot exp(-H/H_0))$$

$$+ \alpha_p \Omega_p\cdot(1+\gamma_1I_1+\gamma_2I_2+\gamma_3I_3)\cdot(\delta_{2018}I_{2018}+...+ \delta_{2023}I_{2023}) + \Sigma\ \alpha_i\Omega_i + \varepsilon \qquad (3)$$

where $I_1, I_2, I_3$ are indicator function for Friday, Saturday and Sunday, respectively, $\gamma_1$, $\gamma_2$, $\gamma_3$ are the corresponding regression

model parameters; $I_{2018}$ - $I_{2023}$ are indicator functions for years 2018-2023 (e.g., $I_{2018}$ =1 for 2018 measurements and 0 otherwise); and $\delta_{2018}$- $\delta_{2023}$ are the corresponding regression parameters. As in case of Model 2, the fitting was done separately for each month of the year, but unlike Model 2, it can be used to estimate urban emissions for individual years.

We used these three different statistical models instead of one model that included all possible parameters to reduce the number of parameters of the models and their uncertainties. That was particularly important for cities with relatively small

urban emissions. As discussed below in Sect. 4, emissions from Model 2 demonstrated that the weekend effect is relatively small in the industrial and background components, that led to the introduction of Model 3.

Note that the plume functions $\Omega$ used in Eqs. (1)-(3) depend on the effective lifetime $\tau$. While $\tau$ can be estimated from the observations, this would lead to larger uncertainties of all estimated parameters. Instead, we used a prescribed value of $\tau$ based on a parameterization for seasonal and latitudinal dependence of $\tau$ that was derived from results of the study by Lange

et al., 2022: $\tau = 1.8 \times exp\ (\ lat\ /\ v)$, where $\tau$ is in hours, $lat$ is the absolute value of the latitude in degrees and $v = 40, 50, 55, 50$ for winter, spring, summer, and fall respectively. This parameterisation gives $\tau$ values of 1.8 h at the equator and 5.5-8 h at 60° depending on the season. The parameterization suggests a slightly stronger increase in $\tau$ with the latitude than in (Lange et al., 2022), but it better matches the estimates for cities in USA and China ($\tau$ =3.9 h for May-September) by (Liu et al., 2016). It should be noted that the uncertainty of $\tau$ estimates and the scattering of $\tau$ values as a function of the latitude is rather large.

Moreover, the lifetime may be changing over time (Laughner and Cohen, 2019) as $NO_2$ concentration declines, although other studies suggest that such changes are minor (Stavrakou et al., 2020). Liu et al., 2016, also found that the average $NO_2$ lifetime



for power plants (3.5 h) may be slightly shorter than for cities (3.9 h). We neglected that possible difference and used the same parameterization for the lifetime for industrial and urban sources.

TROPOMI measurements and all the coefficient estimates in Eqs. (1)-(3) as well as emission calculations are performed for $NO_2$, while actual emissions are in the form of $NO_x$. For $NO_x$ emission estimates, the ratio between $NO_x$ and $NO_2$ is required. Beirle et al., (2021) suggested an algorithm to estimate the $NO_x$ to $NO_2$ ratio for different parts of the world and found that the ratio is about 1.4 over the U.S. and typically between 1.2 and 1.6 elsewhere. The $NO_2$ to $NO_x$ ratio was derived from surface temperature, surface ozone mixing ratio, and the solar zenith angle. We also applied the same upscaling algorithm and some of the $NO_x$ emission estimates are presented in Appendix A.

## 4 Emission estimates for individual urban areas and large regions

### 4.1 Case studies

As an illustration for the method results, **Fig. 1** shows mean TROPOMI tropospheric $NO_2$, the fitting results and three components for Houston, USA, and Guangzhou, China, for four seasons. The average characteristics of the background, urban, and industrial component were analysed using Model 2 that did not contain any year-specific terms. The estimates were done for each month of the year and the results are presented as seasonal values that were calculated as inverse-variance weighted averages of monthly values. For both cities, the fitting results can explain the complex $NO_2$ distribution seen on the TROPOMI map rather well. The residuals are larger for Guangzhou than for Houston due to larger absolute $NO_2$ values. For Houston, the background component reflects the gradient in $NO_2$ that is caused by higher $NO_2$ concentrations over the land and lower over the Gulf of Mexico. For Guangzhou, the background component reflects uneven terrain in the area. There is a clear annual cycle on all three components for both cities. The values are higher in winter and lower in summer that can be, at least, partially explained by a longer lifetime in winter and shorter in summer. From **Fig. 1**, it appears that the changes in the urban component are the largest among all the three components, so they may be caused by a difference in emissions themselves.

### 4.2 Background levels

Model 1 was used to estimate the background component and urban emissions per capita. The estimates were done for individual seasons from June-August 2018 and to September-November 2023. To illustrate the distribution of the background component for different urban areas, **Fig. 2** shows the mean seasonal $NO_2$ values of that component or, in other words, the mean value of $\alpha_0 + (\beta_0 + \beta_1(\theta - \theta_0) + \beta_2(\varphi - \varphi_0))\cdot exp(-H/H_0))$ in DU. There are seasonal patterns in the background component: winter values are higher than summer values at the middle and high latitudes. The background component is particularly large (0.1—0.2 DU or $2.7\times10^{15}$—$5.4\times10^{15}$ molec $cm^{-2}$) in winter over Central Europe and China. Background values over India and China are higher than these over other regions at the same latitudes. In contrast, the background values over Ulaanbaatar, located in central Mongolia far away from large industrial regions, are lower than over Europe and North America at the same latitudes. Another region of high background values is southern Africa, where high values are caused by





biomass burning that is particularly extensive in June-October, i.e., austral winter and spring, although the background values in other African regions and seasons without major biomass burning are low. Low background values can be also seen over South America, Australia, and New Zealand. Not surprisingly, these results suggest that high background values are related to the areas of large anthropogenic emissions.

Individual cities were grouped into 14 large geographical regions with 10–20 areas in each: the US and Canada, Europe-1 and Europe-2, China, India, South-East Asia (also included Pakistan and Bangladesh), the Japan region (that also included Taiwan and South Korea), northern Eurasia (former USSR countries and Mongolia), the Middle East, South and North Africa, Australia and New Zealand, Central America, and South America. Johannesburg (South Africa) and Pyongyang (North Korea) were not included in any particular region because their $NO_2$ emissions were very different from those from neighbouring countries and therefore may bias regional statistics. Note, that there are two regions comprised of mostly

European Union (EU) countries. Europe (EU)-1 includes Italy, France, Spain, Portugal, Belgium, Ireland, and the UK, and EU-2 includes all other countries EU countries plus a few other European cities. These two regions demonstrated very different changes in $NO_2$ during the COVID-19 lockdown (Barré et al., 2021; Fioletov et al., 2022) and we decided to analyse them separately.

Regional mean background values are shown in **Fig. 3a**. The regional means further demonstrated that the highest background values were over China followed by Europe, the Japan region, and India. The background levels over the two European regions are very similar. The lowest background values can be seen in Australia and New Zealand, South America, and summer and fall seasons in South Africa. The spread of the mean background values between the regions is rather large: for example, the background levels over China are 3 times as high as over South America.

**4.3 Emissions per capita**

As mentioned, the urban emission-related term is based on the assumption that every population grid cell's emission is proportional to the cell population only. Therefore, it is natural to express urban emissions as emission per capita and that also makes it possible to compare emissions in different regions.

Estimated seasonal urban $NO_2$ emissions per capita for individual cities are shown in **Fig. 4** and the regional means

in **Fig. 3b**. The highest emissions per capita can be seen over the Middle East, while the lowest occur in India and South Africa (except Johannesburg). The average annual emissions per capita are typically between 7 and 3 kg y$^{-1}$ and lower, about 2kg y$^{-1}$ for India and South Africa. There is also a clear seasonal cycle with a maximum in winter and a minimum in summer. Recall that the urban emission rate is $E_p = \alpha_p/\tau$ and $\tau$ is larger in winter than in summer. Therefore, the seasonal cycle in the total mass $\alpha_p$ is even greater than in the emission rate.

Urban emissions and background component often demonstrate very different seasonal behaviour. For example, the background component for Australia and New Zealand is almost the same in all seasons, while the estimated winter emissions are twice that of summer. The opposite is true for South Africa, where background values for winter and spring are more than twice larger than these for summer and fall due to biomass burning, while urban emissions do not show a large seasonal



dependence. Thus, although the background component is affected by urban emissions, these two characteristics are not identical.

## 5 The weekend effect

Terms responsible for the differences between NO₂ on workdays and weekends were included in Model 2 to study

the weekend effect in the urban, background, and industrial components separately. The results for the background and urban components are shown in **Fig. 5-8**. Most countries have a 5-day workweek with Saturday and Sunday as a two-day weekend, although many Muslim countries have a two-day weekend on Friday and Saturday. Lower, compared to workdays, NO₂ values on Fridays were reported for Saudi Arabia (Butenhoff et al., 2015), Egypt (Khoder, 2009), and Iran (Yousefian et al., 2020). As **Fig. 5a** shows, a significant difference between the urban component values on Fridays vs. workdays take place in the

countries where Friday is a rest day. We grouped all such countries in a special region and excluded them from all other regions when regional averages were calculated for **Fig. 7**. On average, for the region where Friday is a rest day, the difference is about 30% with 1-σ uncertainly of about 5% for all seasons (**Fig. 7a**). For all other regions, the difference between workdays and Fridays is not significant.

For Sundays, the difference with workdays is about 40%-50% for most of the developed countries and between 20%

and 40% for the others except for the region with the rest day on Friday and China. The absence of the difference for the former region was expected. The lack of a weekend effect in China has been noticed before (Kuerban et al., 2020). Beirle et al. (2003) demonstrated the absence of the weekend effect in China and attributed it to the domination of emissions from power plants and industry in total NO₂ emissions. In our study, however, such emissions are largely separated from the urban emissions. Moreover, if industrial emissions are misinterpreted as urban emissions, this would inflate the emissions per capita. As **Fig.**

**3b** shows, Chinese emissions per capita are not very different from those in other regions. More likely, the lack of the difference is related to the traffic control measures, such as the license plate rationing system: First introduced in Beijing in the late 2000s, such measures on workdays are allowing cars that have an even last number of their license plates to be able to drive on roads one day while the cars that have an odd last number of their license plates could go on the road the next day. The rule is now implemented in many Chinese cities including Beijing, Chengdu, Changchun, Shanghai, Guangzhou, Nanchang, and Wuhan

used in our study (https://www.beijingesc.com/news/84-beijing-traffic-control-2012-tail-number-limits.html accessed on March 12, 2024). Such measures cut the number of cars on roads by as much as a half on workdays, but they do not limit the number of cars on weekends. Moreover, for major Chinese cities, trucks are not allowed to entre the cities during daytime on workdays to reduce traffic congestion (Wang et al., 2023). No weekend effect was also found in North Korea.

The difference between emission estimates for workdays and Saturday, in general, shows the same pattern as for

Sunday, although with a lower amplitude. The difference is about 15%-30% for the developed countries and under 20% for the rest of the world. Saturdays are not always rest days India and the difference is not significant there. Saturday is also a rest





day in many Muslim countries and the difference for the region with Friday as a rest day is significant, although not very large, under 10%.

The difference between workdays and weekends for the background component is very different from that for the urban component. It is under 10% and not significant almost everywhere (**Fig. 8**). The only exception are the cities in the Canada and USA region, Europe, and the Japan region, where we can see difference greater than 10% in winter-spring season. These regions and seasons correspond to the areas of the highest background component values (**Fig. 2**). However, even in these regions, the difference is nearly zero in summer. This may suggest that even large changes, up to 50%, in urban emissions over a time period of one-two days lead to a rather weak response in the background component even over the areas of its highest levels.

The industrial component is not in the focus of this study. The study only includes industrial sources in the vicinity of major cities, the number of such sources and their emissions varies greatly from city to city and do not represent total industrial emissions for a particular country or large region. The plot of the difference between the industrial components on workdays and weekend is included in the Appendix B. It does show smaller emissions on Sundays in most of the regions, although the annual mean difference is typically under 20% and the uncertainties are large. For the region, where Friday is a rest day, the difference is about 15%, i.e., about a half of that for the urban component.

## 6 Changes in NOₓ urban emissions

The 5–6-year period of available TROPOMI observations is still too short to accurately estimate long-term trends, but the plots of $NO_2$ emission changes over different regions during the TROPOMI period can be used to illustrate some regional differences in $NO_2$ assuming that the lifetime is not changing with time. Two statistical models, Model 1 and Model 3, were used for such estimates. Model 1 can be used to estimate urban and industrial emissions as well as the background component for each season in 2018-2023. While the model was used to estimate the components for every city, in this study we focus on major features of the $NO_2$ distribution and results for large regions are presented here. Estimates for seasonal (3-month period) were averaged for large regions and the standard errors of these averages were calculated. **Fig. 9** shows the background $NO_2$ estimates for the 14 regions described in Section 4.2.

There are clear differences between seasons and between regions as already shown in **Fig. 3a.** However, the background component does not demonstrate any substantial changes during the analyzed period. Year-to-year variability is typically within the uncertainties with no clear trend. Note that the proxy functions for the urban and background components still can be correlated, that leads to correlation of the parameter estimates: the urban component could be overestimated, and, at the same time, the background component would be underestimated and vice versa. That makes the interpretation of results more difficult: changes in the urban component could be partially attributed to the background component changes. To avoid that problem, Model 3 was introduced. Model 3 assumes that there are no changes in the background and industrial emissions over time. This can potentially affect the urban emission estimates in the opposite side: if there are changes in background or



industrial components, they could be misinterpreted as changes in the urban component. Model 3 also takes the weekend effect into accounts that reduces the uncertainties of the parameters. We found that both models give very similar results. Changes in urban emissions are shown for Model 1, while the Model 3 results are available from Appendix C.

Urban emissions per capita estimated using Model 1 for 14 regions in different years and seasons are shown in **Fig. 10**. For the Middle East, the region with the highest emissions per capita, there is an overall increase in emissions. A positive trend in that region was also noticed by Goldberg et al. (2021b). The lowest emissions were seen in the spring and summer of 2020, and they are likely related to the emissions decline due to the COVID-19 restrictions. The region with the second highest emissions per capita, Australia and New Zealand, demonstrates the opposite tendency: emissions there tend to decline in all seasons. The COVID-19 restriction apparently had some impact on urban emissions during the austral fall of 2020 and the values were lower than in the springs of 2019 and 2021, although 2022-2023 values were even lower. For the Canada and the US region and for northern Eurasia, there is no clear trend in all seasons. The 2020 values were the lowest in spring and summer for both regions.

Both European regions demonstrate nearly the same urban emission values and declining tendencies in all seasons except winter. Note that winter values have high uncertainties due to unfavorable measurement conditions and limited number of high-quality data. The spring 2020 values were about 40% and 20% lower than these 2019 for Europe -1 and -2 regions. They were also lower than the 2021 values. It is interesting to note that springtime 2023 values for Europe-2 were even lower than the 2020 values. The Japan region also demonstrated some decline in urban emissions. The 2020 springtime values were also lower than these in 2019 and 2021. A decline in spring, summer, and fall can be also seen in China. In winter, the 2020 value are clearly an outlier that is almost 40% lower than the values in 2019 and 2021. It is probably a result of the COVID-19 restrictions that started in January (Tian et al., 2020). All other regions do not demonstrate any clear tendencies in urban emissions. One interesting feature is a very large, about 50%, decline in urban emissions from India in 2020 that is probably also related to the COVID-19 restrictions.

**7 Discussion and conclusion**

Contribution from industrial sources, urban areas, and background levels to the observed satellite tropospheric NO$_2$ VCDs were studied using statistical regression models. Three models were developed to analyse these tree components, asses the weekend impact on each of them, and look at the changes during 2018-2023. The analysis was done for 261 major 3° by 4° urban areas around the world grouped into 14 large regions. The statistical models and estimation algorithms were based on our previous study (Fioletov et al., 2022). The background component was considered as a function of the elevation with assumption of a linear gradient in tropospheric NO$_2$ VCD within the analyzed 3° by 4° areas. It was assumed that urban emission depend on the population density and each population grid cell was considered as an emissions source with the emissions proportional the cell population. The industrial component was calculated as emissions from point sources with known locations.



The Middle East is the region of largest emissions per capita. In summer, they are nearly twice of these from, for example, Europe. Moreover, there are indications that the emissions are growing there. It is the only region, where a clear overall increase in emissions occurred. It is possible that per capita emissions for some cities are overestimated if some industrial sources are located in populated areas, however high per capita values for many cities in that region makes this

explanation unlikely. It is also possible that population density data in that region are not as accurate as, e.g., over Europe. We saw some examples that the population density data did not exactly match the maps of populated areas. However, again, it is not likely that such problem exists for multiple cities located in different countries. Also, the upward trend in emissions per capita in that region and a downward trend or an absence of any trend elsewhere may suggest that per capita emissions in the Middle East would likely surpass emissions in other regions.

Emissions per capita for most regions show a clear annual cycle with the highest values in winter and nearly twice lower in summer. The average annual emissions per capita are typically between 7 and 3 kg y$^{-1}$ and lower, about 2 kg y$^{-1}$ for India and South Africa. Emissions per capita and their changes over time in two European subregions are very similar probably because they were the subject of the same EU regulations. The spring 2020 (the COVID-19 restriction period) values appeared as an outlier with about 40% lower emission than in the previous year for Europe-1 and about 20%- lower for Europe-2. The

2023 spring values in Europe were almost as low as the 2020 values in the absence of any restriction measures probably due to an overall long-term decline in emissions.

TROPOMI data demonstrate a strong weekend effect in urban emissions. For cities where Sunday is a rest day, the average difference between Sunday and workday values are 40%-50% for developed countries and 20%-40% for other countries. China is an exception since it does not show any weekend effect. It may be related to the license plate rationing

system that substantially decreases the number of cars on the streets on workdays. For countries where Friday is a rest day, the difference is about 30%. The difference between emission estimates for workdays and Saturday, in general, shows the same pattern as for Sunday, although the difference is smaller, 15%-30% for the developed countries and under 20% for the rest of the world. The industrial component only included industrial sources in the vicinity of major cities and did not represent total industrial emissions for a particular country or large region. It shows similar patterns as the urban component, although with

about twice lower amplitude.

This study further highlights the importance of the background component isolation for studies that use satellite tropospheric NO$_2$ VCD data. The distribution of background NO$_2$ within the analysed areas is modulated by the terrain. The background component includes NO$_2$ from natural sources such as lightning, soil emissions, and wildfires (Hudman et al., 2012; Sha et al., 2021; Zhang et al., 2012) as well as NO$_2$ from anthropogenic emissions that is not directly associated with

urban and industrial plumes. The largest background values are seen over the regions of very high anthropogenic emissions such as China and parts of Europe. Nevertheless, the background component is very different from the urban component. The background component does not show the weekend effect, except for Sundays in winter-spring season for some regions in mid- and high latitudes but even there the difference is 3-4 times smaller than for the urban component. Note that the uncertainties of the difference estimates in winter are large due to unfavourable measurement conditions. A much smaller



amplitude of the weekend effect for the background component suggests, that large abrupt short-term changes in urban emissions on the time scale of 1-2 days does not immediately affect the background component. It means that the $NO_2$ lifetime in the background component is much longer than the lifetime in the urban and industrial plumes that is only a few hours.

The approach described in this study can be used to decompose the observed monthly or seasonal TROPOMI
measurements into three components and then analyse them separately. For example, to study industrial sources, the two other components can be removed from TROPOMI data. Once the statistical model parameters are estimated, the background component can be calculated using elevation data and the urban component can be calculated using population density and wind data.  Then these components subtracted from the original TROPOMI measurements. This would make the plumes from remaining (industrial) sources more pronounced and other conventional methods (e.g. Beirle et al., 2019; Ding et al., 2020;
McLinden et al., 2016) can be used, to detect the source locations. Long-term trends of impact of the meteorological conditions could also be studied for individual components to make the result interpretation easier. Estimates of background $NO_2$ and emissions per capita for large regions can be also used as a benchmark. Individual cities can be compared to their benchmarks to determine if $NO_2$ levels for a particular city are different from average conditions of the region.

**Data availability**

The TROPOMI $NO_2$ product is publicly available on the Copernicus Sentinel-5P data hub (https://dataspace.copernicus.eu, accessed 25 June 2024). The $NO_2$ data used were PAL, v 2.3.1 (until July 2022) and offline mode (OFFL) v2.4 and 2.5 (end of July 2022 - November 2023) were used. The Gridded Population of the World (GPW) dataset is available from NASA Socioeconomic Data and Applications Center at https://sedac.ciesin.columbia.edu/data/collection/gpw-v4 (accessed 25 June 2024). The European Centre for Medium-Range
Weather Forecasts (ECMWF) ERA5 reanalysis data are available from https://cds.climate.copernicus.eu/cdsapp#!/dataset/reanalysis-era5-complete (accessed 25 June 2024). Elevation data are from gridded global relief ETOPO2v2 database (https://www.ngdc.noaa.gov/mgg/global/etopo2.html, accessed 25 June 2024).

**Author contributions**

VF analysed the data and prepared the paper with input and critical feedback from all the co-authors. CM and DG generated
the TROPOMI data subsets for the analysis. XZ performed calculations for the NOx upscaling. HE provided the TROPOMI $NO_2$ data product and related information. All authors read and agreed on the published version of the paper.

**Competing interests**

The authors declare that they have no conflict of interest.

**Acknowledgments.** Sentinel-5 Precursor is a European Space Agency (ESA) mission on behalf of the European Commission. The TROPOMI payload is a joint development by ESA and the Netherlands Space Office. The Sentinel-5 Precursor



ground segment development has been funded by ESA and with national contributions from the Netherlands, Germany, and Belgium. Sentinel 5 Precursor TROPOMI Level 2 product is developed with funding from the Netherlands Space Office (NSO) and processed with funding from the European Space Agency (ESA).



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



**Figure 1.** Mean (2018-2023) TROPOMI NO$_2$, the fitting results, and Model 2-based components for four seasons over two urban four areas as indicated. The columns represent mean TROPOMI NO$_2$ VCD values (column **a**), the fitting results (column **b**), the residuals (column **c**) as well as individual components of the fitting: the background (elevation-related) (column **d**), the urban (population-density-related) (column **e**), and the industrial-source-related (column **f**) components.



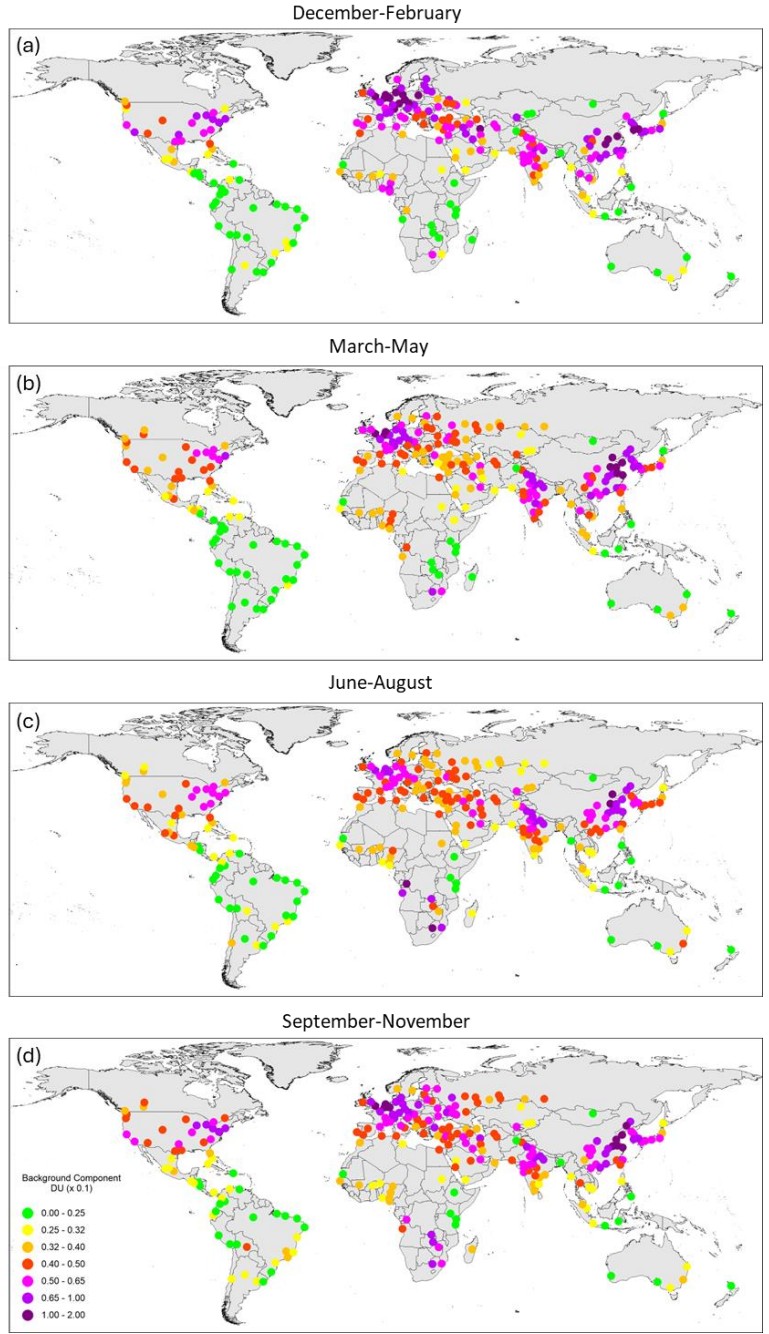

**Figure 2.** The mean NO$_2$ background component in DU ($\times$0.1) for individual cities for (**a**) December-February, (**b**) March-May, (**c**) June-August, and (**d**) September-November. The background component represents the mean tropospheric NO$_2$ VCD when a direct contribution from urban and industrial emissions is removed.





**Figure 3. (a)** The mean background component for 14 regions for four seasons. Mean values for each region were calculated as a mean of the values from all areas for that region. **(b)** The mean values of urban emissions per capita for the same regions. The uncertainty (σ) was calculated as a standard error of the mean. The error bars represent 2σ intervals.



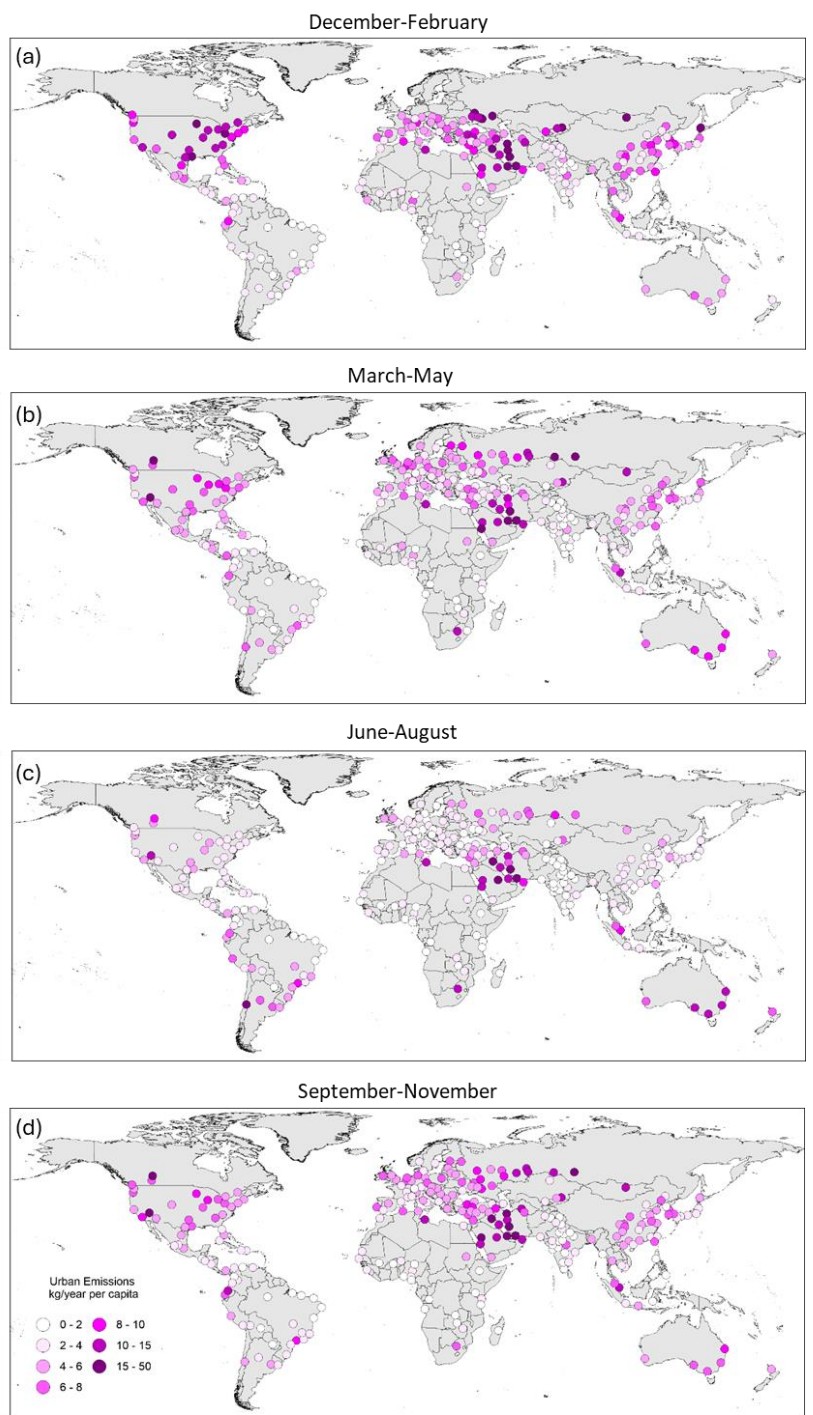

**Figure 4.** NO$_2$ emissions in kt y$^{-1}$ per capita estimated from the urban component for (**a**) December-February, (**b**) March-May, (**c**) June-August, and (**d**) September-November.





**Figure 5.** The percentage difference between urban components on Monday-Thursday and those on (**a**) Friday, (**b**) Saturday, and (**c**) Sunday. Cities where the difference is below 3-sigma level are marked by large gray dots and between 3- and 5-sigma level by small gray dots. Countries where Sunday is a rest day are in yellow, countries where Friday is a rest day are in green.



## Background Component

### Friday

(a)

### Saturday

(b)

### Sunday

(c)

**Figure 6**. The percentage difference between background components on Monday-Thursday and those on (**a**) Friday, (**b**) Saturday, and (**c**) Sunday. Cities where the difference is below 3-sigma level are marked by large gray dots and between 3- and 5-sigma level by small gray dots. Countries where Sunday is a rest day are in yellow, countries where Friday is a rest day are in green. Note the difference in the scale with the previous figure.



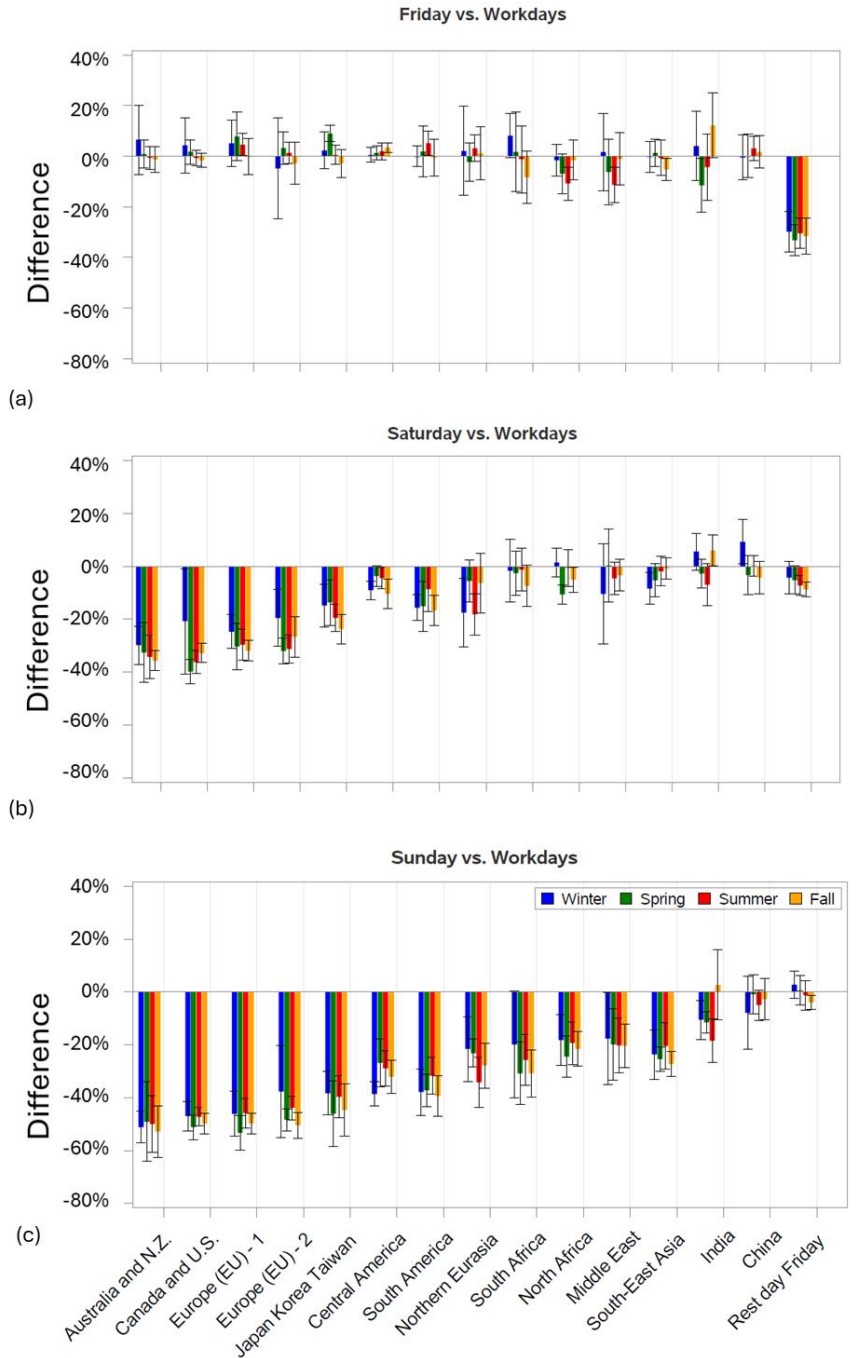

**Figure 7.** The percentage difference between urban components on Monday-Thursday and those on (**a**) Friday, (**b**) Saturday, and (**c**) Sunday for different regions in different seasons. The uncertainty (σ) was calculated as a standard error of the mean. The error bars represent 2σ intervals.





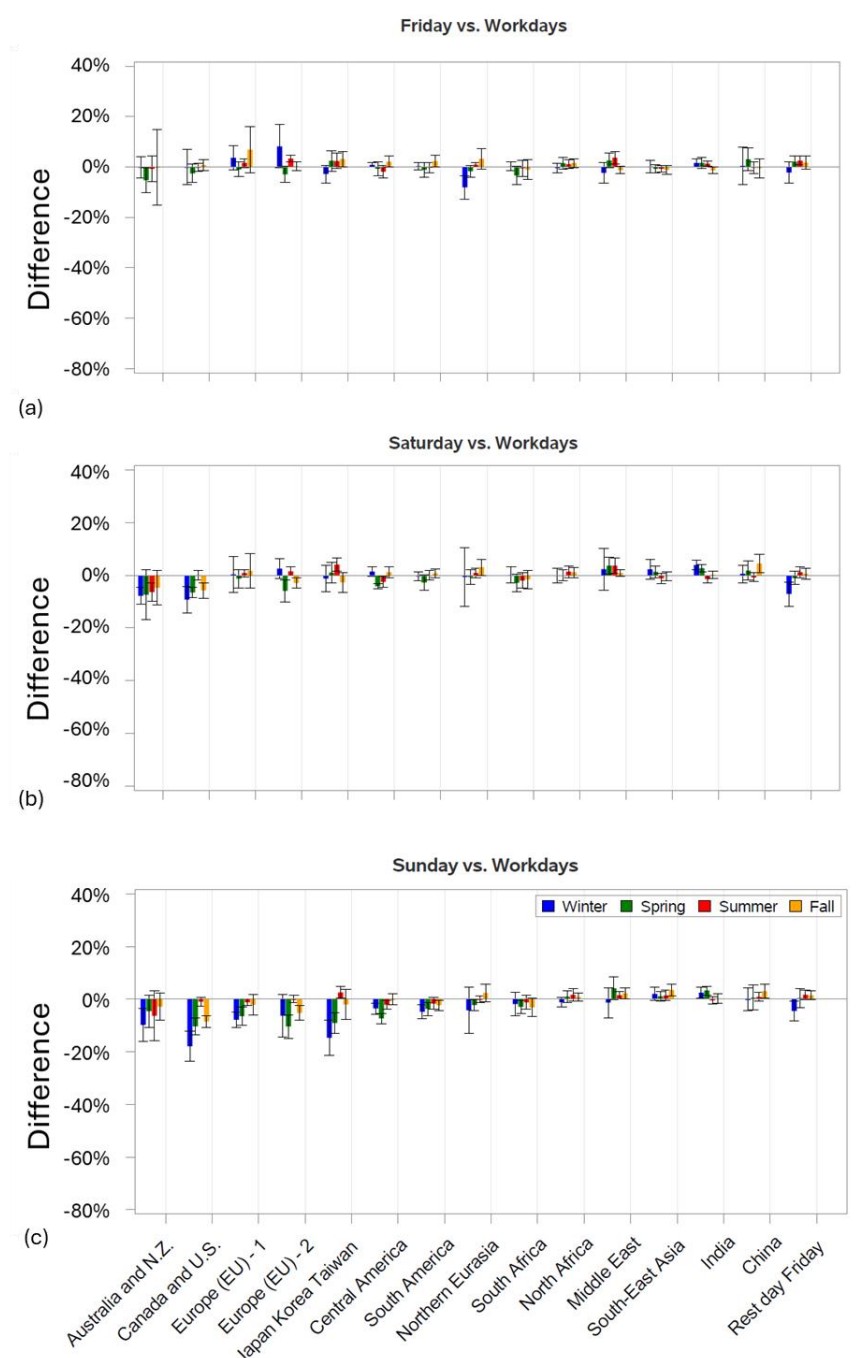

**Figure 8.** The same as Fig.7 but for the background component.



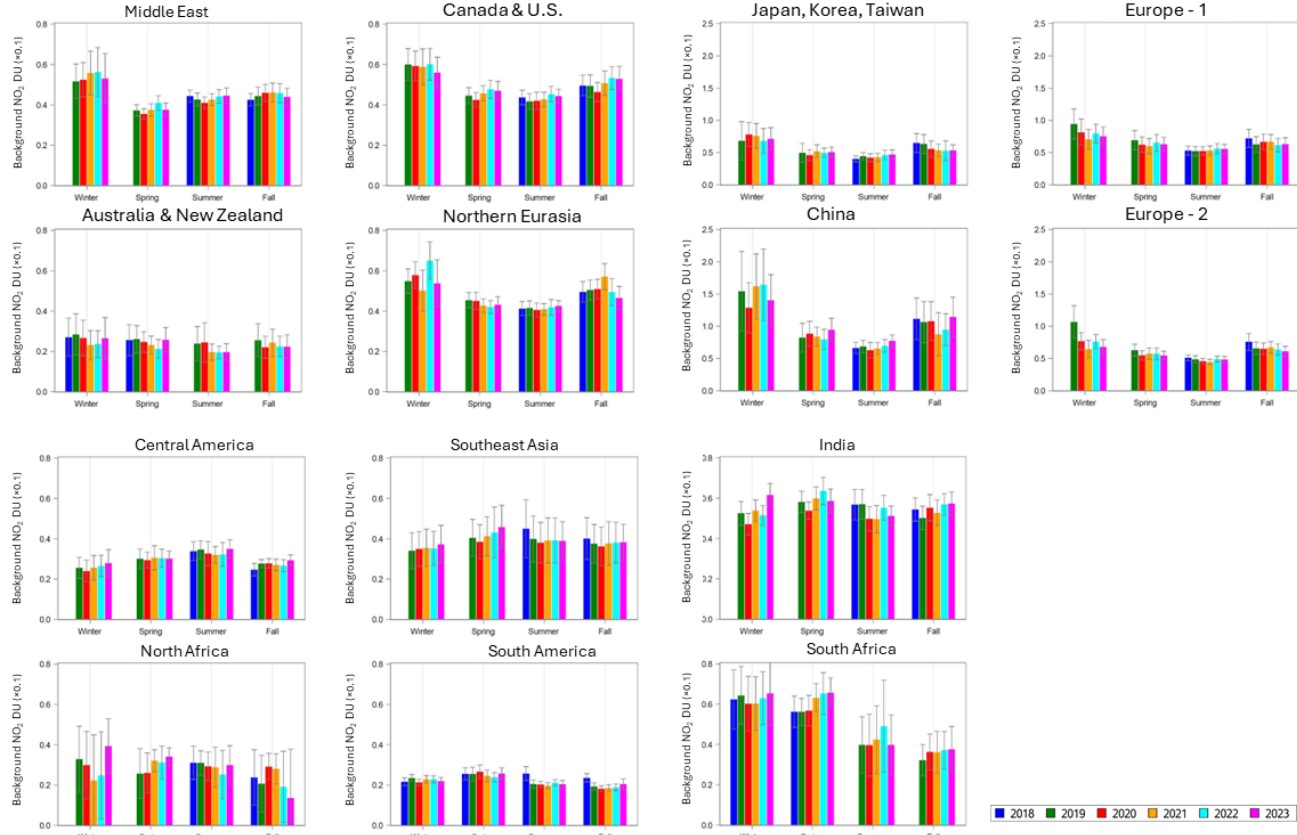

**Figure 9.** The regional mean background component in DU (×0.1), 2018-2023 for individual seasons. Note that the vertical scale on panels for China, the Japanese region, and two European regions is different from the others. The error bars represent 2σ intervals. Also note that different versions of TROPOMI NO₂ data were used (PAL, v 2.3.1 until July 2022 and OFFL v2.4 - v2.5 for late July 2022- November 2023) that may have a small relative bias (see section 2.1).



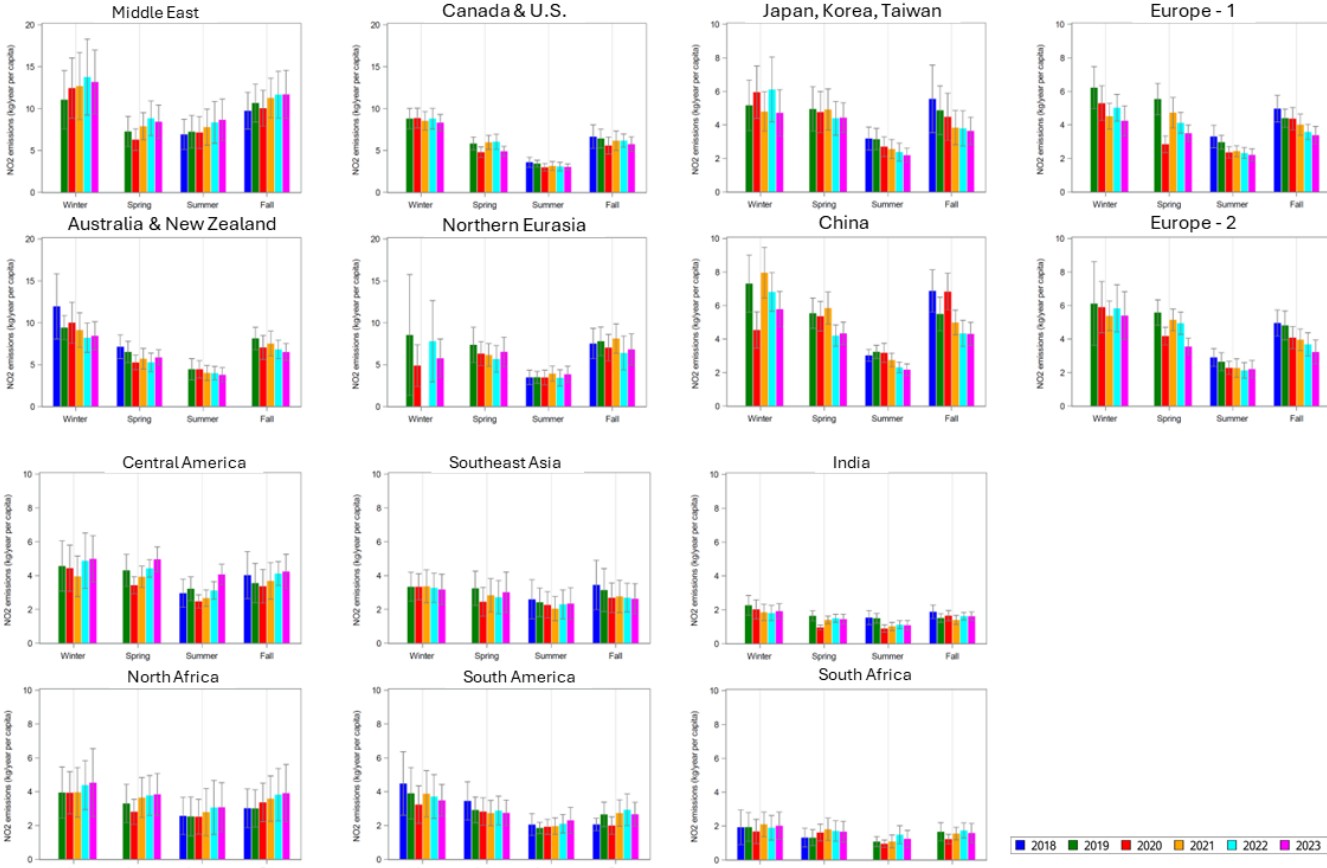

**Figure 10.** The regional mean urban NO$_2$ emissions in kt y$^{-1}$ per capita by region, 2018-2023 for individual seasons estimated using Model 1. Note that the vertical scale on four the top-left panels is different from the others. The error bars represent 2σ intervals. Also note that different versions of TROPOMI NO$_2$ data were used (PAL, v 2.3.1 until July 2022 and OFFL v2.4 - v2.5 for late July 2022- November 2023) that may have a small relative bias (see section 2.1).



**Appendix A: NO₂ to NOₓ upscaling**

Beirle et al., (2021) suggested a way to scale tropospheric $NO_2$ VCD to tropospheric $NO_x$ VCD using temperature and ozone mixing ratio reanalysis data. The same approach was also used by Lange et al., (2022). In this study we estimated potential impact of such $NO_2$ to $NO_x$ upscaling for the calculated emissions per capita.

$$\frac{[NO_x]}{[NO_2]} = 1 + \frac{[NO]}{[NO_2]} = 1 + \frac{J}{k[O_3]}$$

where $J$ is the photolysis frequency and k is the reaction rate constant for the reaction of NO with $O_3$ (Dickerson et al., 1982; Atkinson et al., 2004):

$$J = 0.0167 \times \exp(-0.575/\cos(SZA)) \; (s^{-1});$$

$$k = 2.07 \times 10^{-12} \times \exp(-1400/T) \; (cm^{-3} \, molec^{-1} \, s^{-1})$$

The ozone concentration and temperature profiles as well as surface pressure and geopotential height were obtained from Modern-Era Retrospective analysis for Research and Applications, Version 2 (MERRA-2) reanalysis (https://daac.gsfc.nasa.gov/datasets/M2I3NPASM_5.12.4/summary, accessed 25 June 2024). The reanalysis data has a 3-hour temporal resolution and only the data within SZA<65 degrees are included in the ratio calculation. To match with TROPOMI

15 observations, winter-time data were also been excluded (i.e., for cities that latitude > 40 degrees, December, January, and February data been removed; for cities that latitude <-40 degrees, June, July, and August data been removed). The average estimated ratios are typically between 1.3 and 1.8 as shown in **Fig. A1** with some seasonal dependence. Thus, the $NO_x$ emissions per capita are typically 30%-80% higher than $NO_2$ emissions (**Fig. A2**).

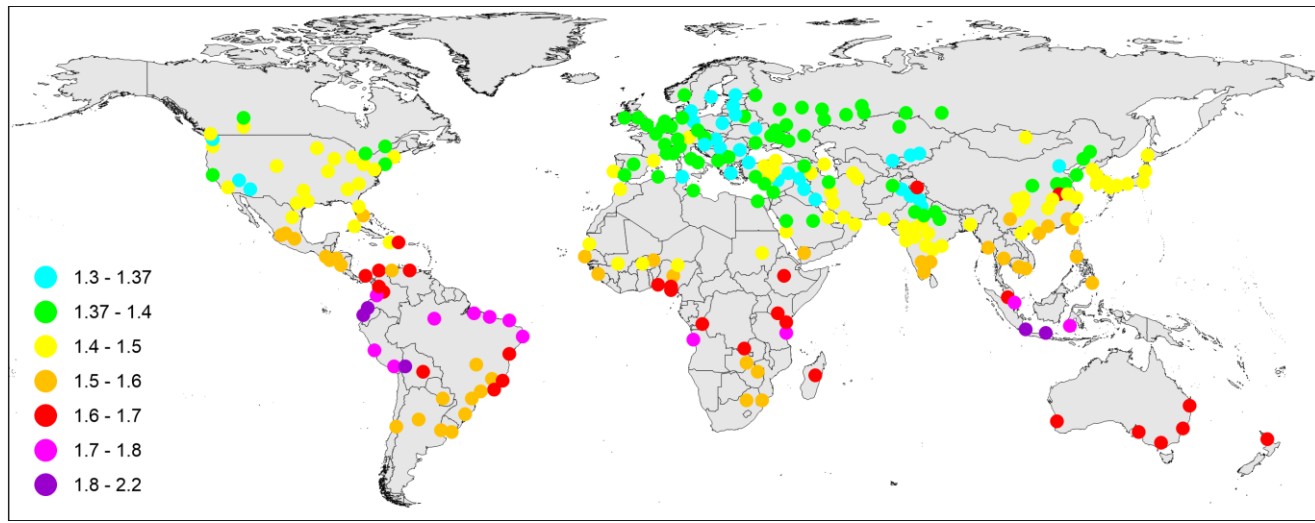

**Figure A1.** The estimated annual mean $NO_x$ to $NO_2$ ratios for the analysed urban areas.





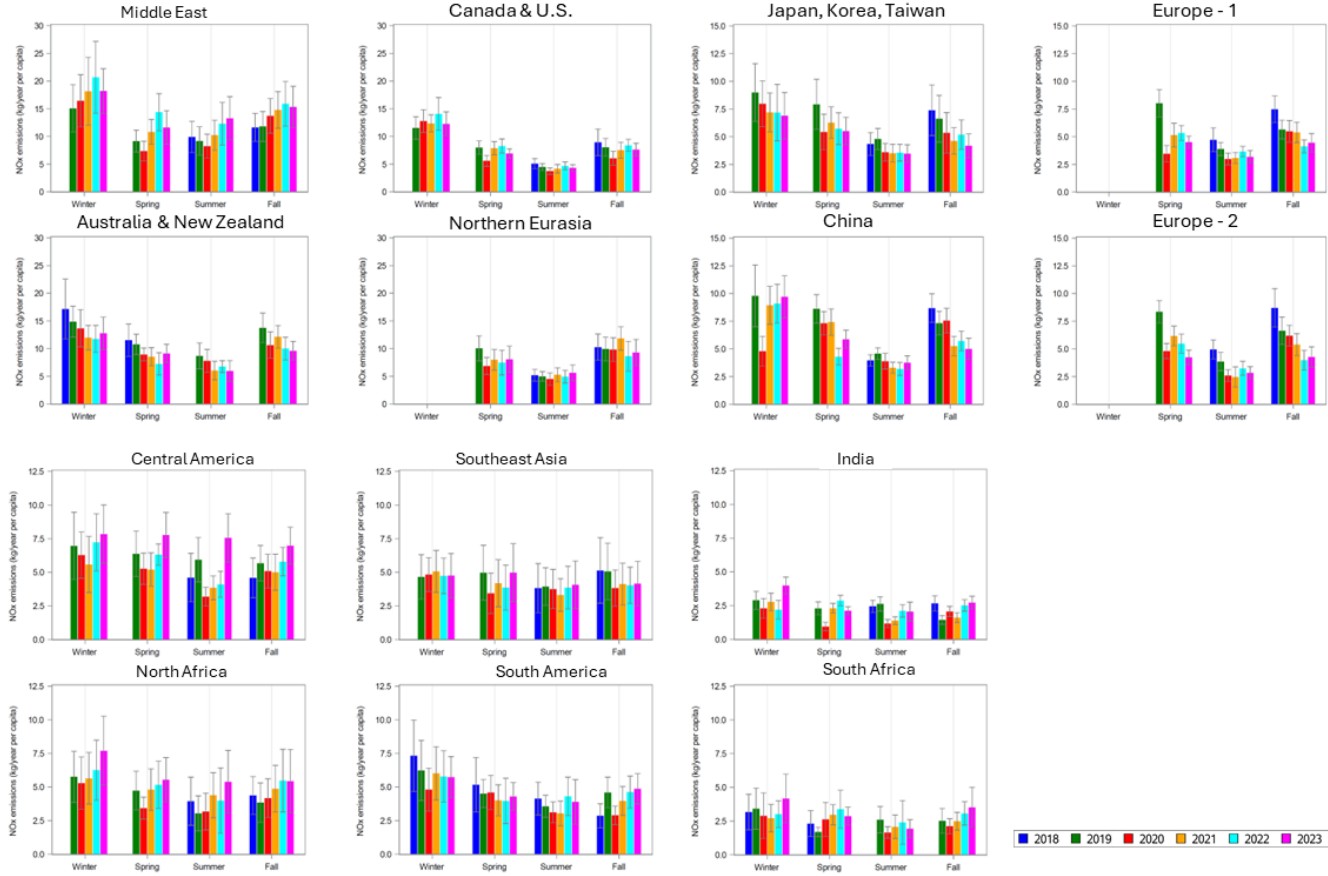

**Figure A2.** Similar to Fig. 10, but for NO$_x$ emissions. The regional mean urban NO$_x$ emissions in kt y$^{-1}$ per capita by region, 2018-2023 for individual seasons estimated using Model 2. The error bars represent 2σ intervals.



**Appendix B: The weekend effect for the industrial component**

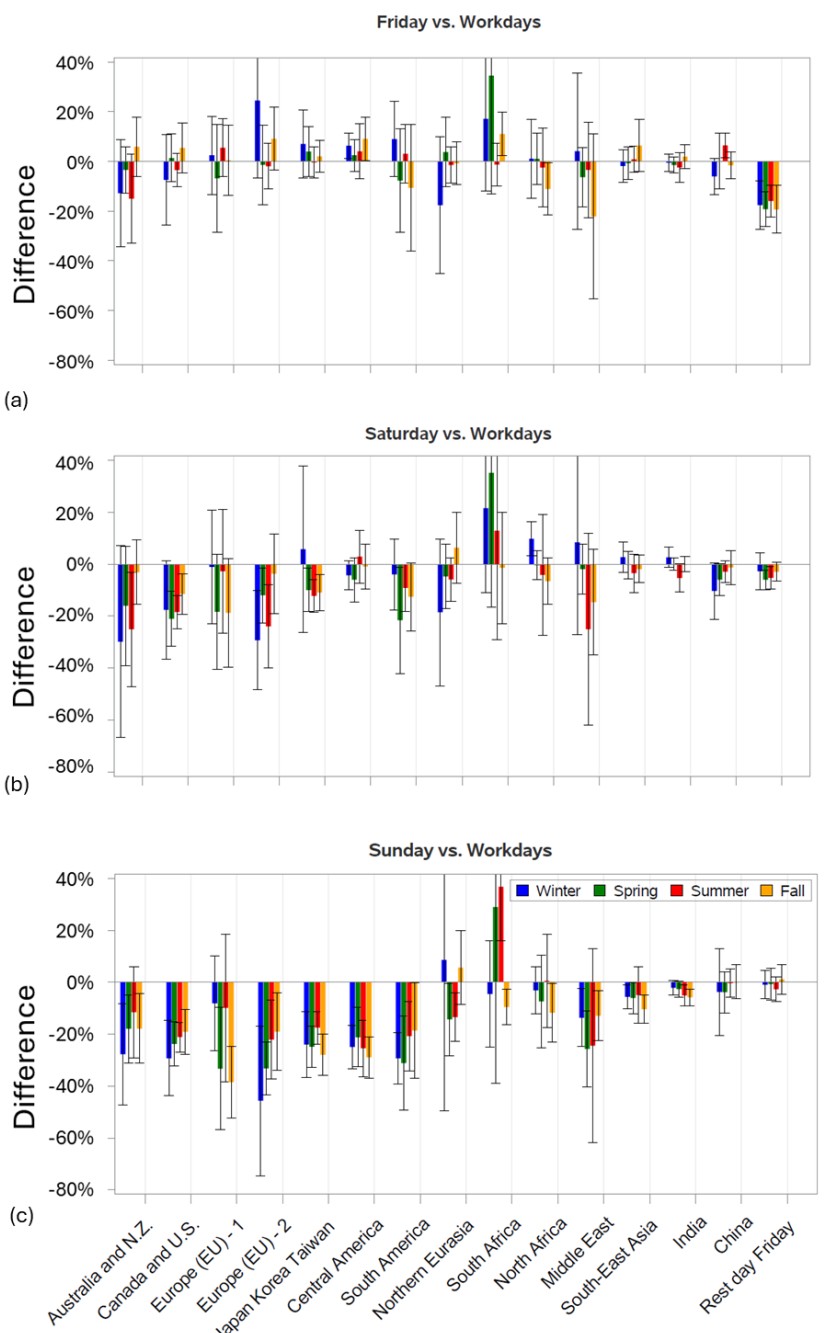

**Figure B1.** The percentage difference between industrial components on Monday-Thursday and those on (**a**) Friday, (**b**) Saturday, and (**c**) Sunday for different regions in different seasons. The uncertainty (σ) was calculated as a standard error of the mean. The error bars represent 2σ intervals.





## Appendix C: Model 3 estimates

Changes in the urban component estimated using Model 1 could be partially attributed to the background and industrial component changes. To avoid that problem, Model 3 was introduced. Unlike Model 1, Model 3 assumes that there are no changes in the background and industrial emissions over time. Thus, all changes over time in $NO_2$ would be attributed to the urban component. However, **Fig. C1** shows that Model 3 results are similar to these for Model 1 (**Fig. 10**). Note that Model 3 also includes terms that accounts for the weekend effect in order to produce a better fit. Model 3 includes the $(1+\gamma_{21}I_1+\gamma_{22}I_2+\gamma_{23}I_3)$ term and the total emissions can be calculated as $\alpha_p\,\Omega_p\cdot(1+(\gamma_{21}+\gamma_{22}+\gamma_{23})/7)$.



**Figure C1.** Similar to Fig. 10, but for Model 3 outputs. The regional mean urban $NO_2$ emissions in kt y$^{-1}$ per capita by region, 2018-2023 for individual seasons estimated using Model 3. Note that the vertical scale on four the top-left panels is different from the others. The error bars represent $2\sigma$ intervals.