# Peer review of "Global seasonal urban, industrial, and background NO2 estimated from TROPOMI satellite observations"

_EGUsphere, 2024_

## Author Comment (AC1)

**Reviewer #1**

Fioletov et al. apply an approach developed in their previous paper (2022, https://doi.org/10.5194/acp-22-4201-2022) to analyze trends in urban NOx emission and background levels in 261 cities around the world. This method was previously focused on the COVID-19 lockdown period, and in this paper is extended to cover the TROPOMI record between 2018 and 2023. The spatial distribution of NO2 observed by TROPOMI is fit using one of three statistical models that include (1) an elevation dependent background term, (2) urban emissions distributed according to population, and (3) industrial emission from zero or more point sources. The three statistical models vary in whether they include terms to account for change per year, day or week differences, or neither. Using these models, they find that (1) urban per capita emissions vary seasonally with largest per capita emissions in the Middle East, (2) there remains a strong weekend (or other rest day) effect visible in the urban emissions, and (3) the background signal does not have a strong day-of-week dependence.

This manuscript is appropriate for publication in ACP. The statistical approach is well thought out and the rationale behind the formulation of each model is clear. The ability to decompose 2D NO2 fields over major urban areas into these three components (population weighted, point source, and background) is valuable to separate different time-dependent behavior from each component, and may be even more so as geostationary observations become generally available. I have two specific questions which should be addressed before final publication, but once those are addressed, I support publication in ACP.

We would like to thank the reviewer for a very thorough and detailed review. We made the corrections suggested by the reviewer.

**Specific comments**

I was surprised by how much structure there was in the background maps for Guangzhou in Fig. 1. Given the form of Eq. 1, where the background is a_0 + (b_0 + b_1(t - t_0) + b_2(p - p_0)) * exp(-H/H0), I was expecting the background to have linear variation along the x- and y-coordinates, more like Fig. 1 shows for Houston. Is the more complicated structure in the Guangzhou background coming from topography? If so, it would be helpful to say so and include a map of surface elevation with Fig 1. If not, then it is crucial to explain where this more complicated structure is coming from, because it appears to have greater NO2 over the city core (based on comparison to column e of Fig 1). That makes me concerned that the background term is somehow taking up urban emissions that don't follow the EMG basis functions represented by Ω_p in Eq. (1); if that's happening, that would mean that the background column amounts and urban emissions cannot be analyzed separately.

Yes, that complicated structure for Guangzhou is largely coming from topography. The topography contribution was discussed in the previous paper (Fioletov et al., 2022, https://doi.org/10.5194/acp-22-4201-2022, their Figure 3. To illustrate this further, we added the

topography and population density maps to the Appendix (new Figure B1). Houston is an example where topography does not play a substantial role because the range of elevations in the Houston area is small, between 0 and 150 m. For the Guangzhou area, the range of elevations is between 0 to 900 m, and it is a case where topography is a substantial factor. In particular, the population density is very low in the northern and western halves of the area and the background $NO_2$ distribution there is comping from topography. We added this information to the manuscript.

My second comment is about Fig. 3. As you point out in the manuscript, emissions derived from fitting EMG functions to observed NO2 data are inversely related to the lifetime (imposed or fit) in the EMG function. That Fig 3b shows a seasonal cycle in almost all regions, with winter emissions greatest and summer emissions least, could also be due to biases in the assumed lifetimes. For example, Kenagy et al. (https://doi.org/10.1029/2018JD028736) saw NOx lifetimes of 29 h (day) and 6.3 h (night) during the WINTER campaign on the US east coast in Feb/Mar 2015, around 40 deg. north. Using the tau = 1.8 exp(lat/v) formula on page 8 gives instead a winter lifetime of 4.9 h. Since $E = a/\tau$, even the shorter lifetime of 6.3 h from Kenagy et al. would reduce the winter emissions estimates by ~30%, which looks like it would eliminate much of the winter/summer difference shown in Fig. 3b. That would be consistent with e.g. Jaeglé et al. 2005 (https://doi.org/10.1039/B502128F), who found that fuel combustion only varies with season in a few regions.

I would suggest that Fig. 3b and other figures that address seasonal variation in emissions incorporate the uncertainty due to lifetime by calculating the emissions with an upper and lower limit for lifetime in each season, in addition to the best estimate of lifetime used currently. Fig. 6d of Lange et al. (2022) shows clearly that several cities' winter lifetimes lie well above their line of best fit (and vice versa in the summer), so the scatter in that figure could be an appropriate source to estimate the uncertainty in lifetime.

Lifetimes or dispersion times for plums in satellite VCD data are not the same as lifetimes in local concentrations. The former are shorter because they reflect not only the chemical transformation, but also the plume dispersion.

Figure 3b shows average values for large regions and therefore it should be appropriate to use average lifetimes. Note that the longest winter lifetimes in Fig. 6d of Lange et al. (2022) are about twice as long as the values of the best fit line. If we use that longest winter lifetime, then indeed winter emissions would be comparable to summer emissions (based on the best estimate of lifetime).

The lifetime and seasonality in emissions issues were also raised by Reviewer #2, who, however, thinks that the seasonal cycle is real and pointed to some studies that suggest that vehicle fuel combustion emissions in winter are higher than in summer (at least, for some types of fuel).

We added some discussion about the lifetime uncertainties, their impact on seasonal emissions to sections 4.3 and 7.

**Minor comments**

- Pg. 2, line 14: confirm that Sentinel-4 is planned for launch this year?

It is now scheduled for 2025. Corrected.

- Pg. 2, line 24: surprised not to see Beirle et al. 2011 (https://doi.org/10.1126/science.1207824) and Valin et al. 2013 (doi.org/10.1002/grl.50267) referenced as, as far as I know, the first papers to apply the EMG fitting approach to NO2 and to combine rotation to align winds with EMG fitting for NO2, respectively.

We added these references

- Pg. 4, lines 13-15, "Since most of our results represent characteristics integrated over 3 by 4 deg areas around major cities, the impact of the change of the version should be smaller than 5-10% step change mentioned above": not necessarily. Changes to the prior profiles' spatial resolution can have a systematic effect on EMG fitting because the higher spatial resolution will often increase NO2 inside a city and decrease it outside. See Laughner, Zare, and Cohen (2016, https://doi.org/10.5194/acp-16-15247-2016) especially Fig. 5.

In the revised version, we used RPRO v2.4 for 2018-2022. So, this problem of different versions is resolved.

- Pg. 5, lines 16-19: would be helpful to give a range for the change in NO2 column per kilometer of elevation, so the reader has a sense of its magnitude compared to a typical background NO2 column amount.

We added a reference to Dang et al., 2023, where the background $NO_2$ vertical profiles are discussed. The decline is roughly 50% per km (Dang et al., 2023, their Figure 2).

- Pg. 6, line 18, "Plumes are described by $\Omega\_p$ and $\Omega\_i$ functions": these are 2D EMG functions, correct? Would be helpful to state that here.

Corrected as suggested

- Pg. 6, lines 19-20, "The plume function for an industrial source has a form...": does the population-related Omega have a different form? If so, what? It might also be helpful to note that these terms will be explained further in later paragraphs - on my first read through, it was a little confusing to go from the plume terms, to the background terms, and back to the plume terms.

We reordered and changed the paragraphs in this section to avoid this confusion.

- Pg. 6, lines 24-25, "the higher is the elevation the lower the background tropospheric NO2 VCD is": might be helpful to refer back to section 2.3 for the physical reason behind this.

We added that reference to section 2.3 and also added a sentence to section 2.3 itself.

- Pg. 7, line 18, "The plume functions $\Omega$ are EMG functions": 1D or 2D? I'm assuming 2D, but if 1D, need to reference how line densities are calculated.

Yes, they are 2D functions. We added that to the text.

- Pg. 8, Eq. (2): several of the $\gamma$ terms have single digit subscripts ("3", "2", and "3") while I terms have subscripts of "13", "32", or "23", which based on the text I'm assuming is a mistake. If not an error, please explain the difference in subscripts between terms.

That was a mistake. Corrected.

- Pg. 8, Eq. (3): would be helpful to note why this model has a different scale factor per year rather than fitting a polynomial or other function that varies over time, since emissions aren't actually step changes at each year boundary. (I'm assuming the scale factors were chosen to avoid assuming a functional form for the emission change over time, but would be good to say so explicitly.)

Yes, it was done to avoid assumptions about a functional form for the emission change over time because such functional form is not known a priory and could be different from area to area.

- Pg. 8, lines 30-31 "...although other studies suggest such changes are minor (Stavrakou et al. 2020).": My understanding of Stavrakou et al. is their conclusions used a model run at 0.5 by 0.5 deg. I'll note that other work (e.g. Valin et al., https://doi.org/10.5194/acp-11-11647-2011) demonstrated that model spatial resolution has a significant impact on the ability to resolve variations in NOx chemistry, which may contribute to the less significant variations in lifetime seen by Stavrakou et al., so perhaps note that the question of how much NOx lifetimes vary over time remains an open question.

Yes, it is indeed an open question since there is no simple way to estimate lifetimes accurately. 1D EMG function-based fits (such as in Valin et al., 2011) work well for point sources, but not for multiple or area sources. The 2D EMG function-based fit can handle multiple and area sources, but it uses prescribed lifetimes. It is possible to test several prescribed values and get the best lifetime value by looking at the residuals, but this approach requires a lot of computational time. We estimated such optimal lifetime for several places and results are similar to these by Lange et al., 2022, although these results are not included in the manuscript.

We added some discussion on this topic and included related uncertainty estimates as described in the Specific comment sections

- Pg. 9, lines 7-8, "We also applied the same upscaling algorithm": please clarify, same algorithm as who or what?

Corrected.

- Pg. 9, line 11, "the fitting results and three components": using which model, model 1?

It is for Model 2. It was mentioned in the figure caption, and we also added that to the text.

- Pg. 10, lines 3-4, "Not surprisingly, these results suggest that high background values are related to the areas of large anthropogenic emission": this statement might need more exploration. Are these areas of high background values also areas where cities and other emissions sources are somewhat close together? If so, then arguably their physical proximity might be playing a larger role in the background NO2 level than the emission source. If, instead, these areas have their emission sources sufficiently separated that one would expect the emission plume from the city to be fully removed through various chemical loss pathways, this statement is somewhat surprising, as it implies that the large emission sources shift the background chemistry in a way that either greatly increases NOx lifetime or shifts the steady-state concentration to a greater amount.

We changed the text to "*the areas of large anthropogenic emissions and dense emission sources*". Indeed, an isolated large emission source or well-separated multiple sources do not change background values very much. We saw such examples in Australia and some other areas.

- Pg. 11, line 27, "...trucks are not allowed to entre": entre -> enter

Corrected

- Pg. 13, lines 2-3, "Changes in urban emissions are shown for Model 1...": shown where/in which figure?

It is in Figure 10. Corrected.

- Pg. 13, line 21, "...a very large, about 50%, decline in urban emissions from India in 2020...": In Fig. 10, it only looks to me like this decrease in present in spring and summer, which would fit the common pattern of the strongest reductions occurring during the first 3-6 months of the pandemic. Perhaps clarify if this statement was meant to apply to only certain seasons.

The reviewer is correct. We added that it was in spring and summer of 2020.

- Pg. 13, line 25, "Three models were developed to analyse these tree components...": tree -> three?

Corrected

- Pg. 14, lines 4-5, "...however high per capita value for many cities in that region makes this explanation unlikely": why does it necessarily follow that seeing high per capita values in many cities makes it unlikely that industrial sources are colocated? Might it represent weaker restrictions on placing industrial buildings in proximity to population concentrations in those regions?

Yes, this is possible in theory. However, we are talking about very large industrial sources (with emissions that are comparable with emissions of a large, >1 million people, city) that are located

in the middle of a city. And, it should be the case for many cities in that region to affect the regional mean. We still think that it is unlikely.

- Fig 1: it might be useful to include maps of elevation and population density for both cities, since these drive the background and urban NO2 emissions.

We added the topography and population density maps to the Appendix (new Figure B1).

- Fig 2, "in DU (* 0.1)": this is ambiguous, does it mean that the actual range of values shown is 0 to 0.2 (i.e. we should multiply the shown values by 1/10) or that it is actually 0 to 20 (i.e. the values were multiplied by 1/10 before plotting)? Would suggest using the values without scaling in the legend.

The values are given in tenths of a Dobson Unit. This this scaling was used to avoid values like 0.025 the legend of Figure 1 that take up too much space. We used the same units for other figures for consistency. In the revised version, we added that the values are given in tenths of a Dobson Unit to Figure 1,2, 3, and 9 captions.

- Figs 5-8: recommend being explicit how percent different is defined, e.g. [(Mon to Thu) - other day]/other day * 100?

The values are given as [(Mon to Thu) - other day]/ [(Mon to Thu)] * 100, i.e. in % of workday values. In the description of Eq 2, it is stated that "$\gamma_{2j}$ are the regression model parameters that represent the departure of Friday's-Sunday's characteristics from these on workdays. In other words, the urban component term is $\alpha_p\Omega_p$ when measurements taken workday are fit and $\alpha_p\Omega_p \cdot (1+\gamma_{21})$ for measurements on Friday (and similar terms for Saturday and Sunday)".

Then, [(Mon to Thu) - other day]/ [(Mon to Thu)] * 100 = $(\alpha_p\Omega_p - \alpha_p \Omega_p \cdot (1+\gamma_{2j}))/\alpha_p\Omega_p$ *100

= $-\gamma_{2j}$*100. We added that the results are shown in percent of weekday values.

- Fig 9: same concern as Fig 2

Corrected as described above for Figure 2.

---

## Author Comment (AC2)

**Reviewer #2**

This is a very well-done manuscript investigating NOx emissions and background NO2 globally. Overall, this was an easy paper to review, so I appreciate the authors producing a well-written manuscript.

We would like to thank the reviewer for a very thorough and detailed review.

"My largest concern is regarding a mixing of two TROPOMI NO2 algorithms v2.3.1 and v.2.4. I realize this does affect any major conclusions of the manuscript, but the current phrasing that "the impact of the change of the version should be smaller than 5-10% step change" is a hypothesis. A change to the surface reflectivity and a priori profiles may not have a large net effect globally, but for some individual cities, this change may be very large, exceeding a 10% difference. An easy, but also appropriate way to deal with this is simply to remove Fall 2022 – Fall 2023 data from the Section 6 analysis. I don't think the paper would be any meaningfully different by doing this. I think it's appropriate to be including the v2.4 2023 data in multi-year averages such as Sections 4 & 5 since those are not investigating interannual trends. If you would like to still include Fall 2022 – Fall 2023 data in Section 6, then I think it's appropriate for you to do a case study… In Page 13 Line 16, you mention that Spring 2023 Europe-2 urban NO2 is lower than Spring 2020. This could be a good test case… see if this still holds true if v.2.4 is used for Spring 2020.

 The revised manuscript is based on versions v2.4 and v2.5 for the entire period. Old version v2.3 was not used anymore. However, the results are nearly the same as in the original version of the manuscript..

Minor suggestions:

Page 3 Line 1 and throughout. Change NO2 emissions to NOx emissions. If necessary (such as on figure captions), rephrase to "NOx emissions reported at NO2".

We changed "$NO_2$ emissions" to "NOx emissions" where appropriate. However, we need to distinguish between emission estimates based on $NO_2$ data (e.g., Fig. 3a, 4, 10) from estimates of NOx emissions (Fig. C1). We also added the clarification "NOx emissions reported at NO2" to the figure captions.

Page 3 (anywhere): It may be good to state explicitly that you are using a variation of an exponentially modified Gaussian function fit, as compared to a variation of the flux divergence method. With one or two sentences, it may be helpful for future readers for you to compare/contrast to the flux divergence methods used by Beirle et al. 2023

 and Lonsdale and Sun 2023 ? I realize your method is discussed in-depth in Section 3, but it could be good to give a quick preview/summary here too.

We added a sentence that "*Unlike other similar studies that studied plumes from emission point sources (e.g., Lange et al. (2022); Beirle et al. 2023), this study included the background component in the analysis and separated urban emissions from emissions from industrial point sources.*"

Page 8 Line 29. The NO2 lifetime is also very strongly a function of ambient VOC and NO2 (Figure 1 Laughner and Cohen), as well as other meteorological factors that are hard to capture such as the amount of vertical mixing / plume width / coastal dynamics. For example, NO2 lifetime in the center of a plume/city is different than at the edges of a plume/city. Liu et al., 2024 gives a list of effective NO2 lifetimes by city (https://acp.copernicus.org/articles/24/3717/2024/acp-24-3717-2024-supplement.pdf) and I don't see a strong dependence on latitude. As a result, I am wary of the Lange et al 2022 parametrization, but also realize it is a step in the right direction. In any case, more discussion of NO2 lifetime uncertainties is needed here.

Yes, $NO_2$ lifetime depends on various factors that could be different from city to city. This is why we also show average results for large regions to establish a "benchmark", so individual cities can be compared to their benchmarks to determine if $NO_2$ levels for a particular city are different from average conditions of the region.

We agree that lifetime in the center of a city could be different than at the edges of a city. However, our isolation of background $NO_2$ should reduce the difference in lifetimes within the plume between the center and the edges of a city.

Liu et al., 2024 analyzed US data from a relatively narrow latitudinal belt, from 25 to 48 N, that may not be enough to see a substantial latitudinal dependence. The absolute values in that study are between 1.8 and 6.8 hours, i.e. in range of Lange et al., (2022) results (their Figure 6). The lifetime estimates by Lonsdale and Sun (2023) also show the latitudinal dependence with lifetime for southern subregion been shorter than for northern subregions. Lonsdale and Sun (2023) also noted that their estimated lifetimes are much longer (from 10 to 100 hours, according to their Figure 1b) than the values from other studies for all regions except southern East Asia. The main problem is that the lifetime estimates based on statistical analysis of satellite data are large for various reasons, not to mention the variability of the lifetime itself.

We added a paragraph on lifetime uncertainty and its impact on emission estimates to section 4.3 and discussion section.

Page 9 Line 11. Can you provide a bit more detail about what the fitting results are actually showing? Essentially this is a statistical re-creation of the TROPOMI NO2? Is that a correct interpretation? Is it simply a summation of background+pop-density sources+industrial sources?

We added a sentence that "*The fitting results (Fig. 1, column b) is a sum of the background, population density-related and industrial source-related components (Fig. 1, columns d, e, f, respectively). The residuals (Fig. 1, column c) are the difference between the mean TROPOMI NO2 and the fitting results.*" We also added Maps of elevation and population density (Fig. D1).

Figure 1. It might be good to annotate percentages on to the panels of these plots comparing the Spring/Summer/Fall values to the winter. For example, annotate that Spring NO2 is 20% (or whatever the exact value) lower than Winter NO2. I think that would strengthen the argument that population density NO2 changes more dramatically than background.

We added such estimates: "*The amplitude of this annual cycle is the smallest for the background component: average summer values are 63% and 76% of average winter values for Houston and Guangzhou, respectively. For the industrial component, these values are 42% and 39%. The seasonal changes in the urban component are the largest among all the three components (summer values are 20% and 32% of wintertime values Houston and Guangzhou, respectively), so they may be caused by a difference in emissions themselves.*"

Page 9 Line 21. I think it'd be OK to further hypothesize why NOx emissions are lower in summer… Evidence suggests vehicle NOx emissions are larger in colder weather than warmer weather and that you may be observing this:
https://www.sciencedirect.com/science/article/abs/pii/S0048969720369333
https://www.sciencedirect.com/science/article/abs/pii/S0269749121016341

Thank you for these references. We added such discussion.

Section 5 (anywhere). It's important to re-emphasize here that TROPOMI is taking an early afternoon snapshot. I imagine NO2 drops in the morning on "rest days" are more dramatic. You could allude to TEMPO / GEMS as next steps to investigate this here or in the Discussion.

We added the following sentence to section 5: "*Please note that TROPOMI takes measurements the early afternoon, and the weekend effect estimates presented in this section may not reflect the difference between workdays and weekends for mornings or evenings.*"

Page 13 Line 16 (also page 14 Line 15). This could be driven by changing algorithm. I recommend doing a case study for this to see if the result holds when using v2.4 for Spring 2023. If so, feel free to keep this sentence. If not, I suggest removal of this sentence.

The revised manuscript is based on v2.4 and v2.5 for the entire period and the Europe-2 springtime 2023 values are still lower than the 2020 values.

Page 15. I was surprised TEMPO / GEMS / Sentinel-4 were not mentioned in the Discussion section. It's important to note that changes in the urban/industrial emissions & background in the morning and late afternoon could be different. This is applicable to all Results sections, but perhaps most applicable to the weekday/weekend section. What new knowledge do you expect TEMPO/GEMS/Sentinel-4 to bring in the context of this work?

This is a good point. We added some discussion about potential impacts of our results on TEMPO / GEMS / Sentinel-4 data analysis.